# Numerical Design of a Thread-Optimized Gripping System for Lap Joint Testing in a Split Hopkinson Apparatus

**DOI:** 10.3390/s23042273

**Published:** 2023-02-17

**Authors:** Bernardo S. Moreira, Paulo D. P. Nunes, Carlos M. da Silva, António Francisco G. Tenreiro, António M. Lopes, Ricardo J. C. Carbas, Eduardo A. S. Marques, Marco P. L. Parente, Lucas F. M. da Silva

**Affiliations:** 1Departamento de Engenharia Mecânica, Faculdade de Engenharia (FEUP), Universidade do Porto, R. Dr. Roberto Frias, 4200-465 Porto, Portugal; 2Instituto de Ciência e Inovação em Engenharia Mecânica e Engenharia Industrial (INEGI), Campus da FEUP, R. Dr. Roberto Frias 400, 4200-465 Porto, Portugal

**Keywords:** impact, high strain rate, Split Hopkinson Bar (SHB), Single Lap Joint (SLJ), thread optimization, adhesive, gripping system

## Abstract

Currently, few experimental methods exist that enable the mechanical characterization of adhesives under high strain rates. One such method is the Split Hopkinson Bar (SHB) test. The mechanical characterization of adhesives is performed using different specimen configurations, such as Single Lap Joint (SLJ) specimens. A gripping system, attached to the bars through threading, was conceived to enable the testing of SLJs. An optimization study for selecting the best thread was performed, analyzing the thread type, the nominal diameter, and the thread pitch. Afterwards, the gripping system geometry was numerically evaluated. The optimal threaded connection for the specimen consists of a trapezoidal thread with a 14 mm diameter and a 2 mm thread pitch. To validate the gripping system, the load–displacement (P−δ) curve of an SLJ, which was simulated as if it were tested on the SHB apparatus, was compared with an analogous curve from a validated drop-weight test numerical model.

## 1. Introduction

Recently, adhesive bonding has been a topic of interest for several industries, given its various advantages, when compared to other joining processes. Adhesive bonding is capable of joining different materials without significant geometric alterations to the components, such as holes for screws or rivets. In this manner, adhesive joining does not significantly add weight to the structure and enables a comparatively more uniform stress distribution in the connection area. Adhesive joints are also characterized by having good fatigue strength, since there are no geometric alterations that create stress concentration. Adhesives also possess high damping properties, which are conferred by the polymeric nature of this type of material [1].

With the increased development and use of structural adhesive joining, the need to mechanically characterize adhesives and adhesive joints has arisen. For structural applications, the determination of mechanical properties becomes essential. Various testing methods have been proposed with the aim of determining the mechanical properties of adhesives under quasi-static loading. However, there are many applications where bonded joints are subjected to impact loads [2]. Consequently, the determination of dynamic mechanical properties is of extreme relevance.

Very few standard procedures exist that measure the dynamic mechanical properties of adhesives. To the best of the authors’ knowledge, the ASTM D950-03 [3] and the ISO 11343:1993 [4] are the only standards for this specific task. The ASTM standardized experimental test consists of impacting the adhesive joint with a pendulum from a pre-determined angle, while the experimental procedure defined in the ISO standard consists of impacting the joint with a weight dropped from a defined height. These standards enable the testing of adhesive joints at higher strain rates when compared to quasi-static tests. However, these strain rates are still considerably lower when compared to other experimental procedures, such as the Split Hopkinson Bar (SHB) test. Consequently, the aim of this paper is to develop tools and procedures to test adhesive joints using an SHB apparatus that, despite not following any standardized procedure, can attain high strain rates [5,6,7].

### 1.1. Split Hopkinson Bar Apparatus Behavior

The SHB experimental apparatus is composed of three main subsystems: the bar setup, the loading device, and the data acquisition system.

The loading device is responsible for the launching of a projectile, referred to as a striker bar or striker tube in many compressive and tensile SHP setups, respectively, generating a stress wave, known as the incident wave (or pulse). The simplest version of a bar setup consists of an incident bar that is hit by the projectile launched by the loading device. With this impact, a stress wave is generated, which will travel through the incident bar (also known as the input bar) until arriving at the specimen. The specimen is fixed between the incident bar and a second bar, which is known as the transmission bar, or as the output bar. Here, the incident pulse will be divided into to a transmitted wave (or pulse), which will propagate through the specimen and the output bar, while the remainder of the incident wave, referred to as a reflected wave (or pulse), will be reflected backwards at the incident-bar–specimen interface.

The acquisition system consists of all of the components used for test monitoring, and must contain, in each bar, two strain gauges in an Wheatstone half-bridge configuration. In this manner, the strain values at the bars are obtained, and, indirectly, the strain and stress levels acting on the specimen can be computed. This acquisition system usually contains other components, such as a striker speed acquisition device [5,6,7]. The test can also be complemented with other technologies to analyze the behavior of the specimen, namely a Digital Image Correlation (DIC) equipment [7,8,9].

Various types of SHB apparatuses exist that are suitable for different states of loading, such as compressive [7,10,11], tensile [5,6,7,12], torsional [7,13], or triaxial loads [7,14], amongst others. While the compressive SHB setup is more commonly used [6,7], this paper focuses on a tensile SHB machine, which is very similar to the typical compressive SHB apparatus. However, there are many different ways to generate a tensile stress wave.

Initial tensile testing setups were actually compression SHB setups with small modifications to enable tensile testing. Harding et al. [15] proposed a bar setup where a hollow weightbar, acting as an input bar, would receive an impact from a striking element, therefore generating a compressive wave. This wave would reflect at the other end, where a yoke is present, thus generating a tensile pulse that partially passes from this yoke to the specimen and, afterwards, to an output bar, both inside of the hollow bar. Lincoln and Yeakley [16] also proposed the use of a compressive setup, where the transmission bar is tubular and of a bigger inner diameter than the incident bar. In this manner, a *hat-type* specimen is placed between both bars, and part of the specimen is tensile tested, despite the fact that the generated stress wave is compressive in nature. Nicholas [17] also proposed the use of a conventional SHB apparatus, with a cylindrical collar between the input and output bars, with a central hole to allow for the presence of the specimen, which is fixed on both bars. As such, a compressive wave is generated, passes through both the specimen and the collar, and is reflected at the free end of the transmission bar as a tensile wave. This tensile pulse then passes exclusively through the specimen, since the collar is not fixed to either bar.

However, the most common tensile SHB setup is the traditional SHB apparatus, with the addition of an anvil at the free end of the incident bar, which is impacted by a striker tube [6,12,18].

The uniaxial stress wave propagation is the main physical phenomenon happening during an SHB experiment. According to the theory of wave propagation, a continuous propagation of the stress pulse is influenced by the cross-section area, *A*, the material density, ρ, and the elastic modulus of the bars, *E*. Therefore, in order to avoid distortion of the stress wave, these properties should remain constant throughout the bar’s length. This statement is true for both the compressive and tensile SHB apparatuses. However, the tensile SHB setup presents another important project requisite. For the compressive apparatus, the tested specimen can either be simply fixed to the bars with the help of friction or placed with the deposition of a thin film of adhesive. However, it is not possible to implement these fixation methods on the tensile SHB apparatus, and they require the use of other accessories [16,17].

### 1.2. Gripping Systems for SHB Setups

The most commonly used procedure to attach the specimen to the bar setup is with the assistance of threaded components. In some cases, the specimen incorporates an external thread, which is fastened to an inner thread present in the bars. For the mechanical characterization of adhesive materials, a design solution was proposed by Nunes et al. [19]. However, when the specimen to be tested has a predefined configuration and cannot feature a threaded edge, a gripping system is required. The development of a gripping system is of major importance for a tensile SHB apparatus since its design will influence the geometric, mass, and elastic variables, thus impacting the wave propagation. Consequently, a wrong design of the gripping system can invalidate the SHB test.

There is some research regarding the design of gripping systems, despite being limited to specific gripping systems. Deng et al. [20] studied the employment of bolts in a gripping apparatus to enable the testing of flat metal specimens and concluded that clamping forces and clearance in the connections significantly alter the stress pulse. A gripping configuration with six bolts is proposed to attain a better accuracy in the stress evaluation of the specimen.

Ganzenmuüller et al. [21] designed a gripping system where the main goal was keeping the mechanical impedance constant until reaching the specimen, thus eliminating unwanted pulse reflections. Given that the gripping system is made of the same material as the bars, both the mass density, ρ, and the Young’s modulus, *E*, are constant. Therefore, the cross-section area, *A*, also remains constant, but the cross-section shape is continuously altered throughout the length. This geometric transition maintains the mechanical impedance of the bar as constant throughout the length of the apparatus. If this could not be maintained in any SHB setup, the stress pulse would be fragmented into various steps. This anomalous behavior is explained in reference [7].

Ledford et al. [22] proposed three alternative gripping solutions: one where the specimen is fixed with adhesive bonding, another one where pins fix the specimen in a similar fashion to the proposal of Deng et al. [20], and a third alternative where the specimen is fixed in a form-fit grip that fastens onto the bars. The specimen fixed in the third gripping system presented a more stable stress–strain curve in the specimen, but also presented the highest drop in strain rate throughout the loading phase. Based on this solution, Constantino [23] presented a similar but improved solution, where the gripping system minimizes the undesirable clamping forces induced by fastening.

As previously mentioned, the most used connection to the bars is through the use of a thread. Given this, Ngueyen et al. [24] studied the influence of threads in the wave propagation, and suggested an optimal design for the threads of specimens or gripping systems.

González-Lezcano et al. [25] performed a numerical study where an initial dynamic-explicit numerical model of a tensile SHB was defined and compared with experimental measurements. Afterwards, in this study, a numerical analysis of the thread clearance was performed to determine if such a parameter would distort the wave. While small clearances did not appear to cause a significant distortion in the stress pulse, one does not understand if the sinusoidal oscillations, traditional of an SHB test, were caused by the clearance or an artifact of the mesh size.

In this paper, the design of a gripping system based on numerical simulations for the exclusive testing of a Single Lap Joint (SLJ) was performed. It is noted that no tensile SHB gripping setup has ever been proposed to test SLJ specimens. A study of various thread parameters—geometric thread shape, nominal thread diameter, and pitch size—was performed. From this study, the optimal thread parameters were defined for the design project of the grips. The proposed gripping system was numerically validated by comparing the obtained load–displacement curves, P−δ, from both an SHB test and a drop-weight test, which was previously validated by Antunes et al. [26].

## 2. Development of the Gripping System

As previously mentioned, the first design requirement to be guaranteed in the development of an SHB gripping system is that the mechanical impedance must remain constant in order to avoid undesirable wave reflections. In other words, the only component that should cause pulse reflection is the specimen itself. In this manner, in order to ensure a constant mechanical impedance throughout the length of the apparatus, the gripping system must maintain homogeneous geometrical, mass, and elastic properties.

Given this, the developed gripping setup was based on the work of Ledford et al. [22] and Constantino [23], and consisted of two grips of 20 mm diameter, as presented in Figure 1. Each grip contains two components: one that is screwed onto either the input or the output bar and contains a cavity where the specimen will be placed, and another part that guarantees the specimen fixture. Both parts are made of X 90 CrMoV 18 stainless steel and are connected with a M4 screw. Note that the specimen will have a special geometric configuration perfectly identical to the aforementioned cavity, thus avoiding an unwanted specimen rupture in this area of the adherend. During the experiment, the gripping system was subjected to an axial load, and, as such, the lower part has a slot with a height and length of 4 mm and 8 mm, respectively, that fits in the upper part to block any axial displacement of the grip. This minimizes the need for excessive tightening force in the screws, which is undesirable for wave propagation [20]. Furthermore, the slot allows for the testing of SLJs with various adhesive and adherend thicknesses; however, to compensate for any remaining space, shims were placed to occupy the remaining space.

This gripping system was mainly designed to test SLJs and adhesive bulk specimens. Both grips are identical and, consequently, one of the grips should be rotated by 180${}^{\circ }$ to fix both edges of the SLJ, as shown in Figure 1. In this manner, one guarantees that the adhesive layer is aligned with the bars’ axis. However, a bulk specimen is not aligned on the central axis, but is parallel to said axis. For the assembly of the bulk specimens, the grips are mounted in the same orientation.

This gripping system is adaptable to a wide range of SLJ sizes. The cavity in the lower part of the grip has a height of 5 mm, thus allowing for the mounting of adherends with different thicknesses, as well as bonded joints with various adhesive layer thicknesses. In each grip, the variation in substrate and adhesive thickness is compensated for with a set of shapes identical to that of the cavity. Therefore, material continuity in the gripping system is maintained.

As previously stated, each grip is screwed to a respective bar. The grip also comprises a slight flat face in the bottom, allowing it to be tightened to the respective bar with a wrench.

Note that the thread is not defined, since it is the study of this work.

## 3. Thread Optimization

The thread shape is an important parameter influencing the wave propagation phenomena [22]. Consequently, a study was made to define the optimal thread shape and size of the gripping system. For this, a calibration specimen was used, similar to the one used by Ledford [22]. This specimen consists of a cylinder with the same diameter as the bar setup, and is threaded on both edges, with the intended thread profile to be analyzed. The objective of such a specimen is to avoid the wave reflection and dispersion characteristic of a normal specimen, and to preserve the wave reflection and dispersion effects occurring due to the threaded connections. Therefore, the effect of the threaded connection in the wave propagation is isolated for a proper evaluation.

In this study, *Abaqus CAE Explicit* finite element method software was used, where two approaches were used. Firstly, a three-dimensional model was initially developed to study the influence that the thread has on the stress pulse propagation. However, given that the required computational cost is extremely high, an alternative using axisymmetric models was performed. In both types of the model, the specimen is mounted in a tensile SHB setup. The input bar, output bar, and striker have lengths of 5700 mm, 2850 mm, and 1500 mm, respectively, while the diameter of the input and output bars is 20 mm. Note that the bars are made of X 90 CrMoV 18 stainless steel, which has a Young’s modulus, *E*, of 210 GPa, a Poisson coefficient, ν, of 0.3, and a mass density, ρ, of 7800 kg m${}^{-3}$. The geometric modeling of the thread in the three-dimensional numerical model is defined according to the geometric dimensions established in international standards and technical literature [27,28,29,30]. However, for the geometric modeling of the threads in the axisymmetric model, some assumptions were taken, which will be further discussed.

To evaluate the impact of the thread in the wave propagation, the stress waves were monitored in locations where a strain gauge would typically be placed in an experimental test. For the particular case of the modeled bar setup, the extensometer in the incident bar was placed at 2400 mm from the interface with the specimen, and the extensometer in the transmission bar was placed at a distance of 330 mm from the specimen. The selection of the location used to monitor the waves has to obey a preset criterion. Commonly, the signal is registered at a certain distance from the specimen location, which assures that the signal that is being obtained is not an overlap of the incident and reflected waves.

The maximum stress in the connection was also evaluated since one of the assumptions of a typical SHB test is that plastic deformation will occur exclusively in the specimen, and that all bars should undergo elastic deformation as the wave is propagating.

### 3.1. Three-Dimensional Dynamic Numerical Model

#### 3.1.1. Model Definition

A three-dimensional thread shape is a complex geometry that can be meshed in any numerical model, thus requiring a very thin mesh size, that consumes a large amount of computational power and takes a long period of time in order to run the simulations. Therefore, a dynamic explicit three-dimensional model, which is presented in Figure 2, was developed to establish a more complete numerical approximation of the real problem at hand, thus establishing a clearer numerical equivalence with the axisymmetric model with regard to the simulation results.

As previously stated, the bars are composed of X 90 CrMoV 18 stainless steel, whose relevant material properties are defined in Section 3. Therefore, all components of the SHB apparatus, including the calibration specimen, were defined in the model with this steel alloy. This is particularly important since the calibration specimen aims to keep the mechanical impedance of the apparatus constant in order to avoid the generation of wave reflections due to the mismatch of the material density or elastic modulus.

As previously stated, the test starts with the impact between the striker tube and the anvil. In this manner, the bars move freely in the axial direction, whereas they remain fixed in the remaining directions, as well as in all three rotations. Furthermore, an initial velocity of 10 m/s in the axial direction was imposed onto the striker tube, since the striker is the only moving element in this direction. Consequently, a load will be generated in this direction and the bars will also move in the same direction. Note that all components of the SHB setup are connected with a tie constraint, thus easing the meshing of each component.

In this model, a hybrid meshing strategy was used. Hexagonal mesh elements were used (C3D8R) for the bar components outside of the area where there is no thread connection. For the specimen and thread regions of the incident and transmission bars, tetrahedral mesh elements were used (C3D4), since it would be impossible to mesh such a complex geometry with hexagonal elements. A general mesh size of 25 mm was defined for the striker tube, with a mesh control of 5 mm both on the inner and outer perimeters of the circular cross-section, and another mesh control of 40 mm in the axial direction. The transmission bar has a mesh size of 1 mm in the threaded area, an elongated mesh size control of 100 mm in the axial direction outside of the zone with the threaded connection, and another mesh size control of 1.6 mm in the circular perimeter of the bar (placed at the free end). The incident bar also has identical mesh sizes for both the threaded zones and non-threaded zones of the bar, but also has a mesh size of 3 mm in the anvil. The calibration specimen has a general mesh size of 1 mm.

#### 3.1.2. Simulation Results

The thread shape tested in the numerical model was the M16 triangular thread shape, with a pitch size of 2 mm. The resultant incident, transmitted, and reflected wave are presented in Figure 3.

The stress distribution can be seen in Figure 4. Some peaks in the stress distribution are noticed; however, these peaks only appear because of excessively distorted elements. These peaks are thus not fully representative of the physical phenomena, and, for the analysis, should not be considered. Most of the load is transferred through the first thread fillets as expected.

### 3.2. Two-Dimensional Dynamic Numerical Model

#### 3.2.1. Model Definition

The axisymmetric numerical model is a two-dimensional geometric model, where a modeled geometry is represented in a given plane. Furthermore, the modeled geometry is symmetrical in relation to an axis and, consequently, this geometry represents a cut view of half of the diametrical cross-section of the real geometry. Given this, some assumptions were taken for this numerical modeling alternative. It is impossible to geometrically represent a threaded helix in a two-dimensional model, which means that the thread fillets need to be simplified and represented as rings with the intended thread profile. As such, it is expected that the evaluated stress values in the thread connection are overestimated.

Firstly, one must consider that the bars are axially loaded, which means that the resistant cross-section area will be in a normal plane of the bar. Given that, in a three-dimensional geometry, the thread describes a helix path, the overall resistant cross-section area differs from the one in a two-dimensional model, where a valley of the female thread decreases the useful resistant cross-section area. Secondly, the load tends to be absorbed in the first fillets of a threaded connection with the ring thread profile configuration. This effect tends to be exacerbated in a two-dimensional geometrical model simulation.

Despite the physical limitations of performing axisymmetric simulations, the goal of these simulations is to make a relative comparison between the thread configurations. This can be achieved, since the expected effects will be present in a comparable manner in all thread configurations.

The axisymmetric numerical model was developed with the initial and boundary conditions as the three-dimensional numerical model, and all components were also defined as being of X 90 CrMoV 18 stainless steel, whose properties are detailed in Section 3. Contact-to-contact was defined in the contacting interface between the anvil and the striker tube, as well as in the interfaces between each bar and the calibration specimen. These contact-to-contact surfaces have a friction coefficient of 0.3.

In the interface between the specimen and each of the bars, the last fillet was not modeled since, for the length of one pitch, the bar and the specimen have the same diameter. Details of the axisymmetric model are represented in Figure 5 and Figure 6.

All of the parts of this model were meshed with a four-node bilinear axisymmetric quadrilateral, with reduced integration (CAX4R). The striker tube has a mesh size of 13 mm, whereas the calibration specimen has a mesh size of 0.5 mm, with a mesh control of 0.2 mm in the threaded area. The threaded areas of both the incident and transmission bars have identical mesh controls. These same bars have a single bias mesh size control varying between a size of 0.5 mm near the thread and a size of 10 mm in the free end of both bars. These bars also have a single bias mesh size control in the radial direction, with a mesh size of 2 mm near the symmetry axis of the bar and a mesh size control of 0.5 mm at the opposite side, where the threaded female connection is located. A mesh size of 4 mm was defined on the anvil, which is placed at the free end of the incident bar.

#### 3.2.2. Comparison with Three-Dimensional Simulation Results

The axisymmetric simulation results were compared with the results from the three-dimensional model by analyzing both the stress waves obtained from both simulations and the stresses occurring in the threaded connection. Note that, in both simulations, a M16 triangular thread was modeled.

In Figure 7a, the input, output, and reflected waves from the axisymmetric model are plotted. Comparing the results from both types of numerical simulations, it can be noticed that the maximum stress wave values from both simulations are identical, yielding, in both cases, an amplitude of approximately 140 MPa.

In Figure 7b, the stress in the thread connection area is represented. As expected, the maximum stress around the first thread fillet, obtained in the axisymmetric simulation results, is considerably higher than that observed in the three-dimensional model simulation. However, in the case of the three-dimensional simulation, the stress wave has a smoother distribution alongside the thread path, and, conversely, in the axisymmetric simulation, the load is almost entirely absorbed by the first fillet.

### 3.3. Thread Comparison Analysis

In this subsection, the thread shapes will be compared, as well as different diameters and different pitch sizes. For this study, the theoretical profile of threads was considered, and, consequently, no clearances were modeled, since typical clearances in such connections are degrees of magnitude smaller than the nominal diameter (maximum clearance with an order of magnitude of $10^{-2}$ mm for the nominal diameters in question).

#### 3.3.1. Thread Shape Comparison

The first parameter to be analyzed using the axisymmetric numeric model methodology is the thread shape. The triangular, trapezoidal, and quadrangular thread shapes were compared. All threads were defined with a 16 mm diameter. These three thread profiles were designed according to their international standards [27,28,29,30].

In Figure 8, the resultant incident and transmitted waves are plotted, which were obtained from simulations where the calibration specimen has a triangular, trapezoidal, or quadrangular thread shape, and a striker speed of 10 m/s.

For all three thread shapes, the obtained behavior is expected, since all stress pulses have trapezoidal shapes, regardless of the simulation where different thread shapes were simulated. If one observes the different reflected wave curves, one can notice that the triangular thread shape presents the biggest reflected wave when compared with the other two thread shapes being studied. The reflected pulses of the trapezoidal and the quadrangular thread shape are almost identical and overlap each other.

The maximum stress in the thread was also computed and is shown in Figure 9 and Figure 10. The three different thread shapes present a maximum stress in the female thread; more specifically, in the first ring of the thread. This is expected since, for a normal thread, the stress is mostly absorbed by the first thread fillets [28]. As previously mentioned, this effect tends to be more acute when the first fillets are represented as rings.

Comparing the stress state in all three female threads, one observes that the quadrangular thread presents a lower maximum stress value. The thread shape where the maximum stress occurs is the triangular shape. However, the trapezoidal thread shape does not provide a large difference in stress behavior when compared with the quadrangular thread shape, thus representing an alternative solution for fixing specimens in an SHB apparatus.

Analyzing the stresses in the male fillets, one notices that the quadrangular thread shape presents a much higher stress value when compared with the other two considered threads. The triangular and trapezoidal threads present a very similar stress value. However, these values are considerably lower than those observed in the female fillet.

Comparing all three options, the trapezoidal thread shape offers the most balanced behavior, with a relatively smaller wave reflection, when compared with the case of triangular-thread-shaped connections, and has an equivalent stress value for the female thread. In terms of wave propagation behavior, the quadrangular thread shape also presents minimal reflection. However, when it comes to stress values, it presents very high values in the male fillets, thus appearing as a worse solution when compared with the trapezoidal thread shape.

#### 3.3.2. Thread Diameter Comparison

After comparing the three thread shapes, the thread diameter was analyzed. The trapezoidal thread was used for comparison. The simulated diameters were 12 mm, 14 mm, and 16 mm, which are established according to the relevant international standards [27,28,29,30].

In Figure 11, the incident, transmitted, and reflected waves of the different thread diameters are represented. The typical trapezoidal waves are clear for all three thread diameters. Comparing the reflected waves, an increase in the maximum stress wave amplitude (in modulus) with the decrease in thread diameter is noticed. The difference between the obtained reflected waves is very low in the cases of the 16 mm and 14 mm threads, but a larger difference is noticed when those two threads are compared with the 12 mm thread, despite still being relatively small.

Table 1 presents the maximum stresses observed in both the female and male thread profiles for trapezoidal threads with nominal diameters of 16 mm, 14 mm, and 12 mm. A clear trend is noticed, where a decrease in stress in the first thread fillet is observed with the decrease in thread diameter. Since the critical area is present in the female thread of the bar, then, as the cross-section area of the threaded part of the bar increases, the stress distribution becomes more uniform, comparatively speaking, and the maximum stress observed decreases.

If one compares the maximum stress values observed in the male thread profile, the trapezoidal 12 mm diameter thread presents a lower stress when compared with the larger sizes. This stress value is comparatively smaller for the M12 thread, but the reflected wave amplitude increases, and, consequently, this case is not an optimal solution. A compromise needs to be made considering the bar material and the desired precision for test measurements. However, it is important to recall that these stress values are not suitable for an absolute comparison given the stated axisymmetric design assumptions.

Another maximum stress component appears in the specimen at the end of the thread shape, which is also indicated in Table 1. It is important to mention that this area of the model is simplified in the design given the stated assumptions for the axisymmetric model. As the sweep path of the thread can be represented, the connection in the thread of the male/female pair is simplified with the internal thread radius. This stress represents the behavior of a common specimen, with a stress concentration at the area where its cross-section is smaller, resulting in possible failure at that zone.

#### 3.3.3. Thread Pitch Size Comparison

The comparison of thread diameter was followed by the analysis of the thread pitch. For the previous simulations, all threads were modeled with a pitch size of 2 mm, which is the standard pitch for the considered thread diameters [27,28]. In this part of the study, the thread pitch sizes of 2 mm, 1.75 mm, and 1.5 mm were compared.

The incident, transmitted, and reflected waves, which are influenced by the different thread pitch sizes, are presented in Figure 12. Once again, all of the measured waves present the typical trapezoidal form. Comparing the reflected waves, which are of equal amplitude, it is concluded that the variation between these thread pitch sizes does not influence the amplitude of the reflected wave.

Table 2 lists the maximum numerically simulated stress values observed in the threads. The 1.5 mm pitch size thread presents a comparatively lower stress, and, once again, this might be justified by the increase in the resistant area in the female thread part. One can see that the stress increases considerably when compared with the TR14×2 trapezoidal thread profile. As such, the TR14×2 thread presents a considerably better performance. However, the selection of the thread pitch also influences the thread helix sweep path alongside the thread length and, consequently, the contact angle of the threaded parts changes in relation to the bar axis. In the axisymmetric model, the variation in this condition cannot be represented. Therefore, for the thread pitch comparison, a three-dimensional analysis would be necessary to extract more concrete conclusions.

Similarly to the thread diameter comparison, a maximum stress value appears on the final part of the thread (where the female thread begins in each of the bars) rather than on the thread fillets. In contrast with the stress of the fillets, the 1.5 mm pitch thread shape presents the higher stress value.

In order to analyze the importance of the stress expressing near to the bar/specimen interface, a similar numerical study was made, but, instead, a regular specimen type, which has a reduced cross-sectional area in the failure zone, was used. The threaded connection of the specimen is identical to the one in the calibration specimen; however, the main section has a diameter of 6 mm. This specimen was modeled using only a linear elastic analysis model, with properties identical to those of the X 90 CrMoV 18 alloy. The area where the thread connection ends was monitored to verify if the high stresses still occur.

In Figure 13, the maximum stress in the mentioned area is plotted. With a decrease in the specimen diameter, the numerical stress values in the stress concentration area diminish in a very significant manner, and the maximum stress remains below 200 MPa. This means that, in the case of a typically tested specimen, this stress does not appear, and thus is not relevant for design purposes.

With this, it is concluded that a 14 mm diameter trapezoidal thread with a pitch size of 2 mm is the optimal thread profile for a bar setup with a diameter of 20 mm.

## 4. Numerical Validation of the Gripping System

The developed gripping system was subjected to a numerical validation. Since the gripping system was designed to test overlap joints, this validation procedures consisted of simulating a test of an SLJ specimen mounted on the gripping system. From the numerical simulation, a load–displacement, P−δ, curve was obtained and compared with the numerically obtained curve from the test of the same SLJ in a drop-weight machine setup. Numerical simulation results from the drop-weight test were used for validation, since the SHB apparatus with the proposed fixture, shown in Figure 2, is still in a design project stage. In this manner, the use of an already validated drop-weight impact numerical model that can recreate equivalent strain rate conditions to those of an SHB apparatus with low striker velocities is required.

For this purpose, the modeled adhesive was the crash-resistant, modified-epoxy Betamate ™120 EU structural adhesive. The static properties of this adhesive were used, which means that the strain rate dependence of the adhesive was not taken into account, since this can be acceptably assumed given the relatively low strain rates achieved with a wave generated from the impact of a striker tube at 3 m/s. The properties of the adhesive are presented in Table 3. The dimensions of the SLJ tested are represented in Figure 14.

The numerical modelling of a drop-weight test allows for a validation of the SHB test simulation, where conditions non-dependent on the test setup are identical. In this manner, for the SHB apparatus model, an SLJ was mounted in the gripping system, which was installed in the same bar setup, with a striker speed of 3 m/s. To create analogous conditions in the drop-weight simulation, an impactor with the same mass as the SHB striker tube was defined, with the same impact speed of 3 m/s. It is noted that a drop-weight test cannot attain high impactor velocities, such as the 10 m/s speed typical of an SHB test.

### 4.1. SHB Numerical Model

To replicate these conditions, an FE model was developed using *ABAQUS Explicit* numerical simulation software. The SLJ, grips, and bar setup were modeled using *SolidWorks*, and were then imported to *ABAQUS*. Consequently, this model is three-dimensional in nature.

In this model, the thread shapes were not considered, which would represent an impractically large computational cost, and were simplified with a cylindrical geometry. This model simplification was carried out since the reflected wave in the thread optimization models is conversably smaller than the transmitted and incident waves. In this manner, while it is true that the thread does influence the wave propagation at the interface, the thread parameter optimization allowed for the minimization of such a phenomenon. The shims, which are used to guarantee that the SLJ specimen remains horizontal and colinear with the loading axis, as shown in Figure 1, were simplified, becoming part of the gripping system body. The numerical model is shown in Figure 14.

The materials used for this model are X 90 CrMoV 18 stainless steel alloy and the Betamate™120 EU adhesive, whose properties are listed in Section 3 and Table 3, respectively.

The adhesive layer properties were established according to the pure mode of triangular cohesive zone model laws. For the damage initiation, the quadratic nominal stress criterion was considered, which can be described by the expression
(1)$${\left(\frac{t_{n}}{t_{n}^{0}}\right)}^{2}+{\left(\frac{t_{s}}{t_{s}^{0}}\right)}^{2}=1$$
where $t_{n}^{0}$ and $t_{s}^{0}$ are the normal cohesive stress and the shear cohesive stress, respectively. These stress components correspond to the normal tensile strength, $\sigma_{max}$, and shear tensile strength of the material, $\tau_{max}$, respectively.

The damage evolution is described by a linear relation and the mode mixity is described by a linear power law, as expressed in the following equation:(2)$$\frac{G_{I}}{G_{IC}}+\frac{G_{II}}{G_{IIC}}=1$$

In this manner, the damage initialization and evolution is described by a mixed-mode triangular law that is dependent on the load components.

The established boundary conditions are identical to the ones in the thread optimization model, except for the striker speed. This time, an initial velocity of 3 m/s was imposed on the striker. Two tie constraints were imposed between the bars and the grips. The adhesive layer was also connected to the adherents through the tie constraint. Since the gripping system is connected to the lower part of the gripping system, a tie constraint was also used in this connection. All other interactions between bodies were defined by a tangential contact interaction behavior without friction, such as the impact of the striker in the bar setup. General contact with default specifications was defined in *ABAQUS*.

This model was meshed with different techniques to obtain an optimal mesh. The bars and striker were meshed with 3D stress eight-node linear brick elements with reduced integration (C3D8R). The grip components, due to their geometrical complexity, were meshed with a combination of 3D stress tetrahedral (C3D10M) and hexahedral elements (C3D8R). This combination was used in the same components and the interface between the two element types was compensated for with a tie constraint. The lap joints were meshed with C3D8R elements for the adherents and a 0.2 mm cohesive layer composed of an eight-node three-dimensional cohesive element (COH3D8). For the bars, a larger mesh size was defined alongside the bars’ direction (150 mm), but with a diametrically refined size of 5 mm, and, for the specimen zone, the mesh was refined to approximately 1 mm.

### 4.2. Drop-Weight Test Numerical Model

In order to validate the results obtained with the SHB model, a comparison was performed with simulation results from an already validated model of the drop-weight test of the SLJ. This model was previously validated by the authors in reference [26].

This is a two-dimensional model of a lap joint that is loaded by an impactor, as is presented in Figure 15, where the model’s boundary conditions are presented, as well as the meshed component. The joint is fixed with an encastre in one of the edges, and the opposite edge is attached with an impactor that has a pre-set mass.

For this model, the substrates were meshed with a four-node bilinear plane stress quadrilateral element with reduced integration (CPS4R). The cohesive layer was meshed with a four-node two-dimensional cohesive element (COH2D4). The mesh was refined in the area of the cohesive zone layer, where the size varied between 0.2 mm and 0.4 mm.

This model was previously used in the literature and its results were compared with the data obtained from an experimental drop-weight test. Machado et al. [2] characterized the behavior of a lap joint bounded with a DP 8005 adhesive from 3M. The experimentally obtained load–displacement curve, P−δ, was compared with the one obtained with this model. The test impact conditions were set, with the 26 kg mass block impacting the joint at 1.75 m/s. These curves are plotted in Figure 16.

The obtained curves using this model were a good prediction of the experimental data. This two-dimensional model has demonstrated to be a useful tool in estimating the behavior of an SLJ, subjected to the impact loading of intermediate impactor velocities.

### 4.3. Result Analysis

For the SHB model, the strain at the bars were monitored. The incident and reflected wave were obtained in the input bar at a distance of 2400 mm from the specimen. The transmitted wave was monitored in the output bar at a distance of 330 mm from the interface with the specimen. The strain signals are shown in Figure 17. The signals present the expected trapezoidal shape, which is characteristic of a stress pulse. In the case where specimen failure happens, there is a difference between the incident and transmitted signal, and also the presence of a very significant reflected wave. The difference between these waves allows for an estimation of the specimen behavior.

With the transmitted strain signal, it is possible to calculate the force experienced by the specimen. Assuming stress equilibrium between the bars and the specimen, the force is computed as
(3)$$P=A_{B}E_{B}\varepsilon_{T}$$
where $A_{B}$ is the bar’s cross-sectional area, $E_{B}$ is the bar’s elastic modulus, and $\varepsilon_{T}$ is the strain measured in the transmission bar due to the transmission wave.

By extracting the displacement at both edges of the lap joint, a load–displacement curve can be plotted. However, it is important to mention that, in order to plot this curve, a time shift must be made. Since the strain signal is read at a certain distance from the specimen, a force calculated with the strain signal corresponds to a displacement that happened before that strain was read. This time interval corresponds to the time taken for the wave to travel from the specimen until it reaches the reading point. Consequently, the time shift can be computed as
(4)$$\Delta t=\frac{\Delta x_{\mathrm{strain}\;\mathrm{read}}}{C_{B}}$$
where $\Delta x_{\mathrm{strain}\;\mathrm{read}}$ is the distance from the specimen to where the strain is read, and $C_{B}$ is the wave propagation speed of the bar material [7]. Since $\Delta x_{\mathrm{strain}\;\mathrm{read}}$ corresponds to 330 mm, the time shift is equivalent to
(5)$$\Delta t=6.3599\cdot 10^{-5}\;\left[s\right]$$

Finally, the P−δ curve is shown in Figure 18, and it is compared with the load–displacement curve from an analogous SLJ tested with the drop-weight apparatus.

Both P−δ curves have a similar shape, with a corresponding maximum load of 4 kN. The energy required to break the lap joint is also very close for both tests, and a difference of 0.1 mm displacement at the rupture point is noticed. The different load conditions of the two simulated tests justify these slight differences. Whereas, for the drop-weight test, the SLJ is fixed at one of the edges and the other edge is loaded, in the numerically simulated SHB test, the SLJ is attached via a clamp in the bar setup, which has an unrestricted axial degree of freedom.

It is also relevant to mention that, for both simulations, the maximum stress in the adhesive is higher than the shear strength, and this is explained by the dynamic inertial effects occurring during the test.

An estimation of the strain rate can be computed. The strain rate of the specimen is obtained as
(6)$$\dot{\varepsilon_{s}}=\frac{2C_{B}}{\varepsilon_{r}l_{s}}$$
where $l_{s}$ is the length of the specimen, $\varepsilon_{r}$ is the reflected strain, and $C_{B}$ is the wave propagation speed in the bar’s material. One must point out that, for the case of an SLJ specimen, the length of the specimen is replaced by the thickness of the adhesive layer, which is 0.2 mm. This is carried out since an SLJ is a multi-material specimen and the strain rate is altered in a significant manner just by considering the material properties of the epoxy adhesive (in essence, a polymeric material) or the properties of the stainless steel alloy. Therefore, the presented strain rate in Figure 19 is considerably overestimated if one uses the theoretical calculation of the strain rate, as presented in Equation (Equation 6). Furthermore, this effect is compounded by the fact that the strain rate is calculated with the measured strain from the reflected wave.

## 5. Conclusions

The tensile SHB test is useful for characterizing materials under high strain rate conditions. However, it presents the difficulty of fixing specimens to the bar setup in a simple manner, especially considering that the specimen can have several geometric configurations.

For the case of the SLJ specimens, a gripping system was designed with the focus of avoiding stress wave reflections before the incident wave reaches the specimen. This proved to be a big challenge, especially when considering that the gripping system needs to have an easy assembly process.

Considering that the gripping system is fixed to the bars through threading, a study for an optimal thread was performed. This study was conducted using an axisymmetric numerical model, which has demonstrated to be a suitable tool for the relative comparison of the different thread geometries. However, it was not useful for as absolute comparison, since the stress values in the female thread are overestimated.

With this study, it was concluded that the optimal thread for a bar setup with a diameter of 20 mm is a trapezoidal thread with diameter of 14 mm and a 2 mm pitch size. Although the referred thread has the best performance for wave propagation, it is also noticeable that the difference between threads does not have a very significant impact on wave propagation, with a striker speed of 10 m/s. Experimental testing should be performed to evaluate the impact of the threads on the wave propagation and, consequently, on the test measurements.

After selecting the optimal thread, the gripping system was validated, irrespective of the thread shape. A model was developed to obtain the load–displacement curve, P−δ, of an SLJ. This curve was compared with the curve obtained with a previously developed numerical model of an impact test in a drop-weight machine. The curves presented an approximate value of energy necessary to break the joint and the maximum force in the SLJ, therefore validating the SHB model and the gripping system.

## Figures and Tables

**Figure 1 sensors-23-02273-f001:**
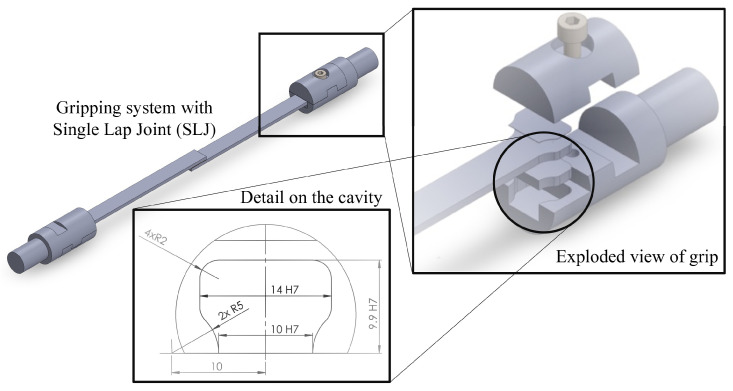
Model of gripping system with a detailed view of the cavity (dimensions in mm).

**Figure 2 sensors-23-02273-f002:**
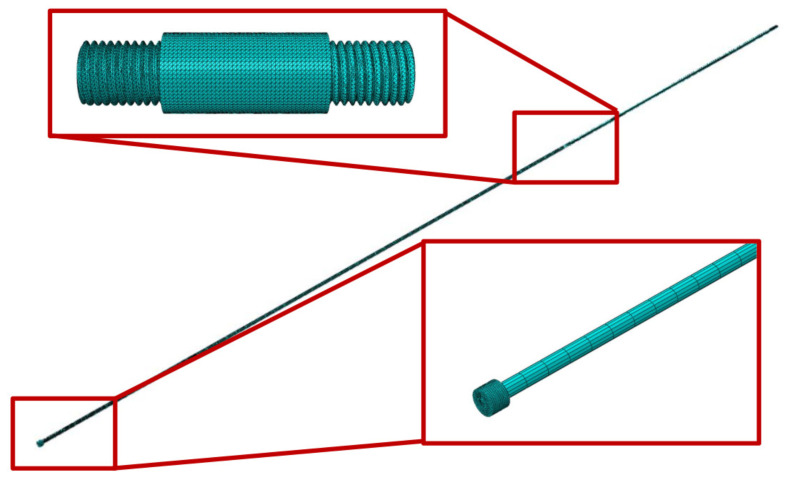
Three-dimensional model of the bar apparatus with the calibration specimen.

**Figure 3 sensors-23-02273-f003:**
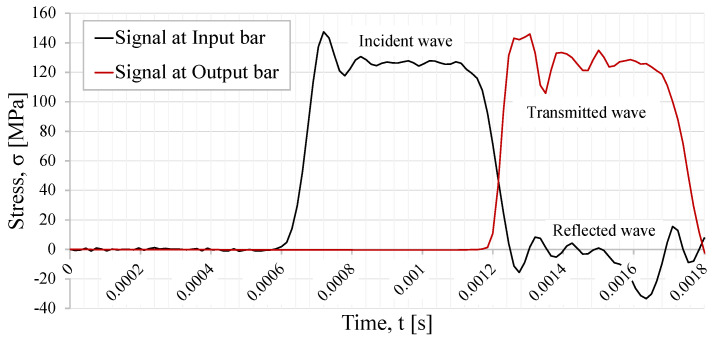
Incident, transmission, and reflected waves from the three-dimensional simulation results.

**Figure 4 sensors-23-02273-f004:**
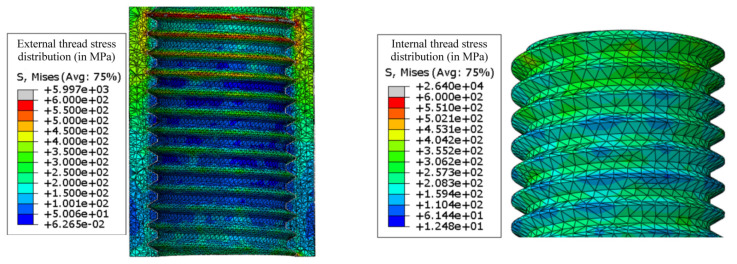
Thread stress distribution in the three-dimensional model.

**Figure 5 sensors-23-02273-f005:**
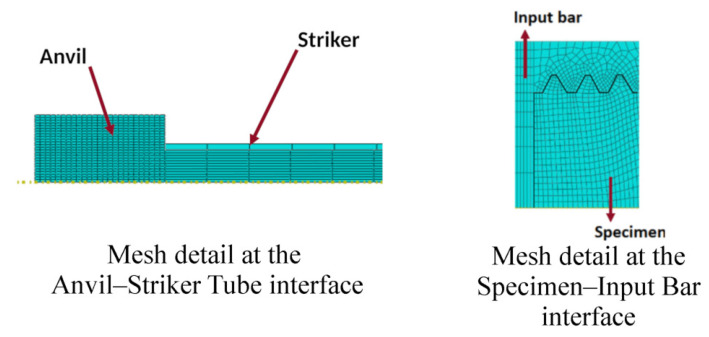
Mesh details for the asymmetric model with a triangular thread shape.

**Figure 6 sensors-23-02273-f006:**
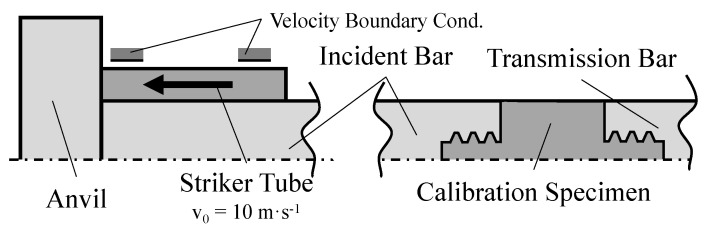
Schematic representation of the model with initial and boundary conditions.

**Figure 7 sensors-23-02273-f007:**
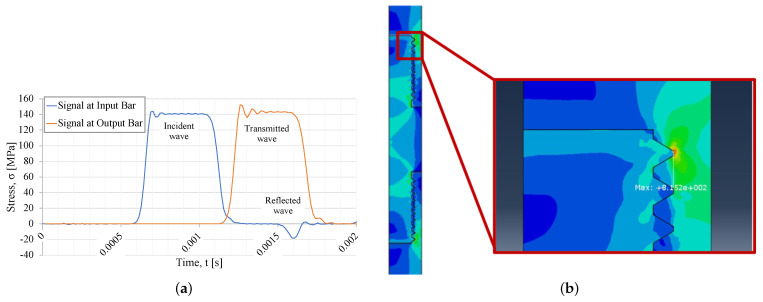
Simulation results from axisymmetric model. (**a**) Stress waves from the axisymmetric simulations. (**b**) Maximum stress in thread (stress in MPa).

**Figure 8 sensors-23-02273-f008:**
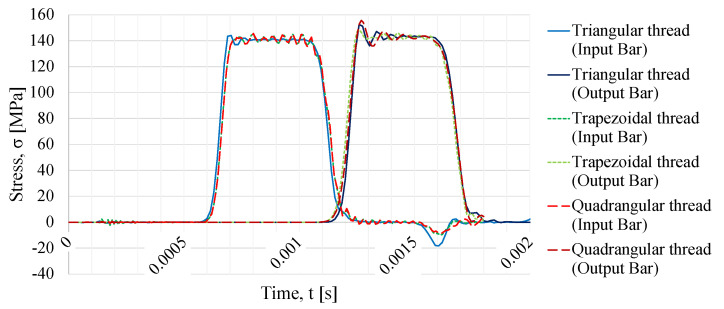
Incident, transmitted, and reflected waves for different thread types.

**Figure 9 sensors-23-02273-f009:**
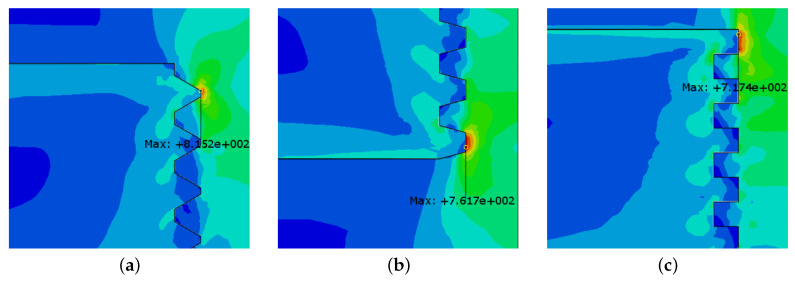
Maximum stress (in MPa) in the thread connection for different thread shapes (female thread). (**a**) Triangular thread. (**b**) Trapezoidal thread. (**c**) Quadrangular thread.

**Figure 10 sensors-23-02273-f010:**
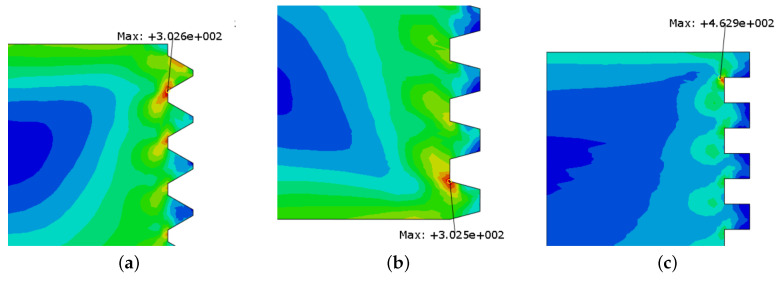
Maximum stress (in MPa) in the thread connection for different thread shapes (male thread). (**a**) Triangular thread. (**b**) Trapezoidal thread. (**c**) Quadrangular thread.

**Figure 11 sensors-23-02273-f011:**
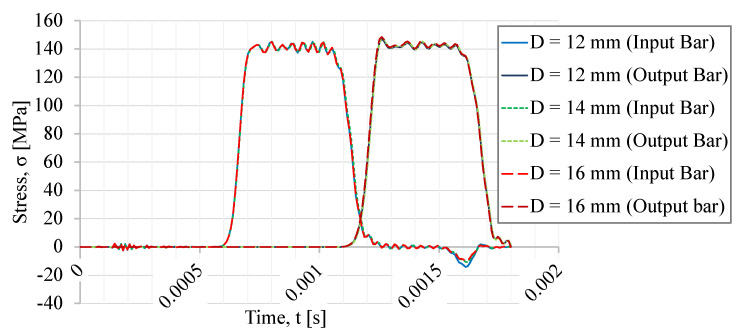
Incident, transmitted, and reflected waves for different thread diameters.

**Figure 12 sensors-23-02273-f012:**
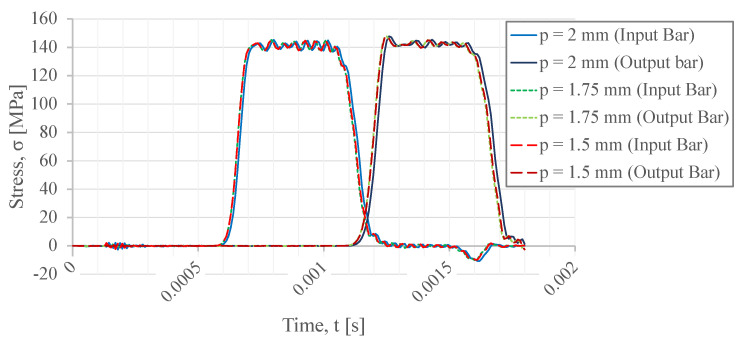
Incident, transmitted, and reflected waves for different thread pitch sizes.

**Figure 13 sensors-23-02273-f013:**
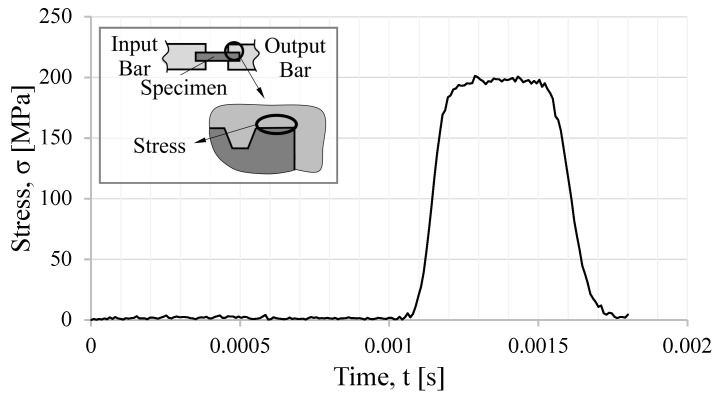
Maximum stress in the neighborhood of the bar edge, as indicated in the legend.

**Figure 14 sensors-23-02273-f014:**
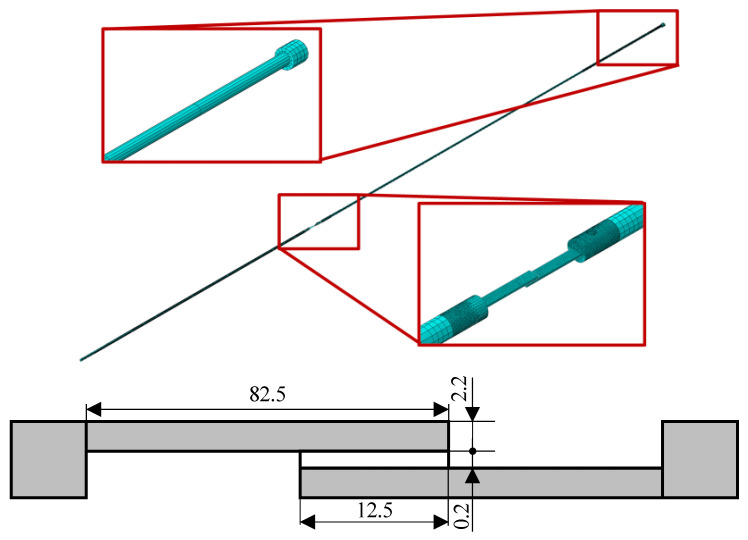
Numerical model of the SHB apparatus with the designed gripping system and a single lap joint specimen (with specimen dimensions in mm) for validation.

**Figure 15 sensors-23-02273-f015:**
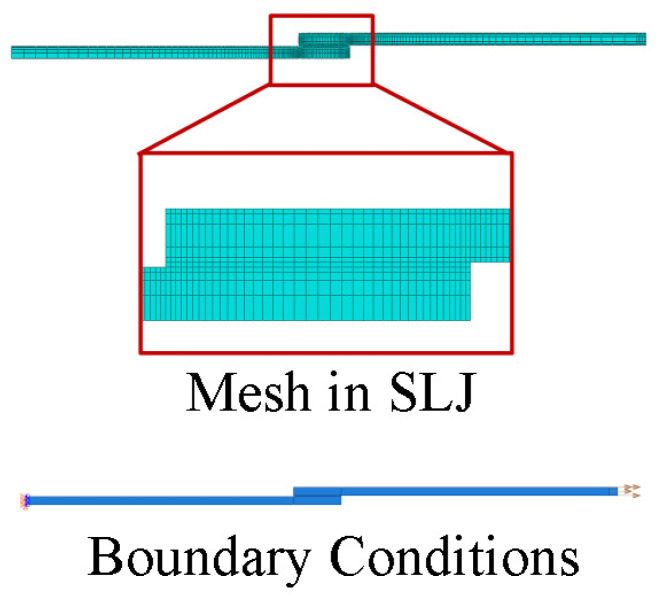
Numerical model of a single lap joint specimen being tested in a drop-weight apparatus.

**Figure 16 sensors-23-02273-f016:**
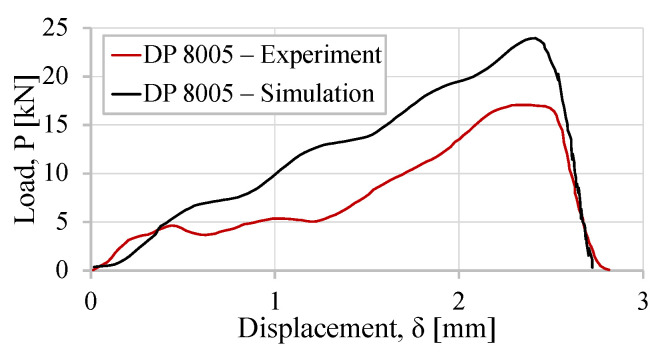
Load–displacement curve, P−δ, for the DP 8005 lap joint.

**Figure 17 sensors-23-02273-f017:**
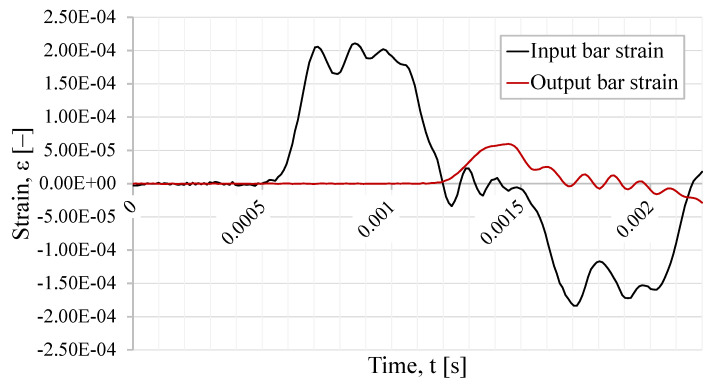
Incident, reflected, and transmitted strain waves from the SHB validation model.

**Figure 18 sensors-23-02273-f018:**
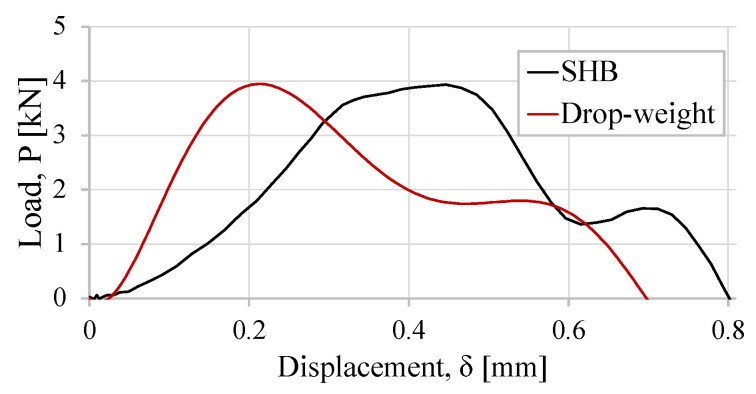
Load–displacement curves (P−δ) of the SLJ with the crash-resistant adhesive for the SHB and the drop-weight test validation models.

**Figure 19 sensors-23-02273-f019:**
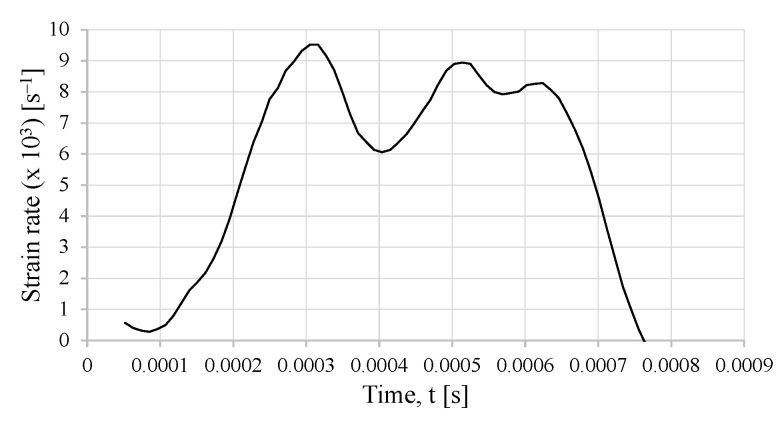
Specimen strain rate estimated by the bar strains with the SHB model.

**Table 1 sensors-23-02273-t001:** Maximum stress (in MPa) in the thread connection for different thread shapes.

	D=12 [mm]	D=14 [mm]	D=16 [mm]
Female thread	488.6	581.7	761.7
Male thread	302.5	283.1	302.5
End of thread (in bars)	817.2	612.8	412.7

**Table 2 sensors-23-02273-t002:** Maximum stress (in MPa) in the thread connection for different pitch sizes of an M14 trapezoidal thread.

	p=1.5 [mm]	p=1.75 [mm]	p=1.75 [mm]
Female thread	603.4	591.6	761.7
Male thread	364.5	346.2	302.5
End of thread (in bars)	696.8	708.2	612.8

**Table 3 sensors-23-02273-t003:** Properties of the Betamate™120 EU adhesive [31].

Tensile strength, $\sigma_{max}$ [MPa]	28.1
Shear strength, $\tau_{max}$ [MPa]	30.9
Elastic modulus, *E* [MPa]	2100
Shear modulus, *G* [MPa]	787.74
Mass density, ρ [g cm${}^{-3}$]	1.26
Fracture toughness for mode I, $G_{IC}$ [N mm${}^{-1}$]	2.5
Fracture toughness for mode II, $G_{IIC}$ [N mm${}^{-1}$]	10.7

## Data Availability

Not applicable.

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
