# Peer review of "Numerical Design of a Thread-Optimized Gripping System for Lap Joint Testing in a Split Hopkinson Apparatus"

_sensors, 2023, doi:10.3390/s23042273_

Round 1

Reviewer 1 Report

Dear Authors, this paper contains information regarding the research in gripping system for lap joint testing.

Brief Summary of the Study

In this study, Mechanical characterization of adhesives is done using different specimen configurations at Single Lap Joint (SLJ) specimens. A gripping system was conceived to enable the testing of SLJs, which is attached to the bars through threading. Also, the gripping system geometry was numerically evaluated.

Evaluation of the Study

From my point of view, the study contains a good idea about gripping system for lap joint testing in a Split Hopkinson apparatus. However, when approached from a scientific point of view, some deficiencies have been seen in the manuscript. My comments and recommendations as follows:

1.    In the study, the problem is well expressed, sufficient literature is given and the purpose of the study is clearly stated.

2.    The literature research is up-to-date and has the quality to express the problem well.

3.   All numerical analyses was described well and results meet each other.

3.    In the manuscript, sections are not meet with the “research manuscript section” of Sensors.

4.   “Material and Method” section and “Result” section should be given separately.

            For example:

            1. Introduction

            2. Material and Methods

                        2.1. Development of the gripping system

                        2.2. Thread optimization

                        (etc.)

            3. Results

            4. Discussion

            5. Conclusions

5. All results should be given in the “Result” section in the scientific paper. In its current situation of manuscript is complicated therefore is been hard to understand. So, authors should create result section and all results should be given under the “Results” section.

6. In the references, 5th and 6th references have same name. It should be checked and corrected.

           References:

         5. Boland, A.J.; Lopes, A.M.; da Silva, C.M.S.M.; Tenreiro, A.F.G.; da Silva, L.F.M.; Nunes, Paulo, D.P.; Marques, E.A.S.; Carbas, R.J.C. Development of a Split Hopkinson Pressure Bar Machine for High Strain Rate Testing of Bonded Joints. Journal of Testing and Evaluation 2022, 50, 20200677. https://doi.org/10.1520/JTE20200677.

6. Tenreiro, A.F.G.; Silva, C.M.; Lopes, A.M.; Nunes, Paulo, D.P.; Carbas, R.J.C.; Silva, L.F.M. Development of a Split Hopkinson Pressure Bar Machine for High Strain Rate Testing of Bonded Joints. Mechanism and Machine Theory 2022, 159, 104289. https: //doi.org/10.1016/j.mechmachtheory.2021.104289.

Author Response

Please find the response in the attached file.

Reviewer 2 Report

The ABSTRACT section is well–structured. The aims and objectives of the research are well-presented.

The paper is structured properly. The paper contains a study about the design of a gripping system, based on numerical simulations, for the testing of adhesive joints SLJ. The proposed gripping system is numerically validated.

The INTRODUCTION section provide the necessary background information. The section is well–documented. It is comprehensive with the state of research in the field, relevant for this subject.

All sections of the paper are relatively well described and include detailed information about the implementation details. Overall, the sections are technically and fairly detailed.

The CONCLUSION section succinctly summarizes the major points of the paper, presented concisely and to the point. The authors fairly conclude in just a few sentences given the rich discussion in the body of the paper. Overall, this section is well-concluded.

The Tables are representative and the Figures & Graphs have good qualities.

The list of REFERENCES is long and relatively well chosen. The entire bibliography is current. I noticed a percentage of about 21% self-citation.

I have not detected any mistakes.

Author Response

(The authors gave the same response as above.)

Reviewer 3 Report

Comments and Suggestions for Authors

Authors have submitted the results of Numerical design of a thread optimised gripping system for lap joint testing in a Split Hopkinson apparatus. Overall, the investigation was performed in a considerable details. But still, the depth of the contents is shallow and the result must be improve. Additionally, some shortcomings are evident as:

1.        The author mentioned in the Abstract that, an optimisation study to select the best thread was performed, and by regarding the documentation of your numerical problem. The documentation and the description of the optimization method is missing. Discus.

2.        In the introduction part, the literature review is very poor and needs improvement and the reorganization of the literature review would be beneficial to the paper.

3.        Authors must mention the novelty of their work in the last paragraph of the introduction.

4.        The enumeration of the paragraphs should be checked. For Example line 132 the word components is repeated and need to be remove.

5.        Table 1. Elastic and mass density properties of the X 90 CrMoV 18 stainless steel. Remove the table and write the properties in the text and add reference.

6.        It is suggested to add the mesh sensitivity analysis results.

7.        In Figure 5. Mesh details for the Asymmetric model with a triangular thread shape. You need to add another picture showing the boundary condition.

8.        Re arrange Figure 3. Incident, transmission and reflected waves from the three-dimensional simulation results and Figure 6. Simulation results from axisymmetric model to be one figure.

9.        Needs to add figure showing the Maximum stress in the thread connection for different thread shapes. Triangular Thread, Trapezoidal Thread and Quadrangular Thread instead of Table 2.

10.    Needs to check Figure 7. Figure 8. Figure 9. To show the Incident wave and Transmitted wave

11.    Minor revision is required in terms of spell checks and grammatical errors

Author Response

(The authors gave the same response as above.)

Round 2

Reviewer 1 Report

I checked the rewised version of the manuscript in detail. I could conclude that this paper can be accepted for publication in its current situation. 

Author Response

(The authors gave the same response as above.)

Reviewer 3 Report

Comments and Suggestions for Authors

Authors have submitted the results of Numerical design of a thread optimised gripping system for lap joint testing in a Split Hopkinson apparatus. Still some shortcomings are evident as:

1.        The literature review still needs some improvement.

2.        Still Minor revision is required in terms of spell checks and grammatical errors

Author Response

(The authors gave the same response as above.)
